# Neural Spatiotemporal Point Processes: Trends and Challenges

**Sumantrak Mukherjee**                                       *sumantrak.mukherjee@dfki.de*
*Data Science and its Applications, DFKI*

**Mouad Elhamdi**                                              *mouad.elhamdi@um6p.ma*
*College of Computing, Mohammed VI Polytechnic University*
*and Data Science and its Applications, DFKI*

**George Mohler**                                              *mohlerg@bc.edu*
*Department of Computer Science, Boston College*

**David A. Selby**                                             *david_antony.selby@dfki.de*
*Data Science and its Applications, DFKI*

**Yao Xie**                                                    *yao.xie@isye.gatech.edu*
*Georgia Institute of Technology*

**Sebastian Vollmer**                                          *sebastian.vollmer@dfki.de*
*Data Science and its Applications, DFKI*

**Gerrit Großmann**                                            *gerrit.grossmann@dfki.de*
*Data Science and its Applications, DFKI*

**Reviewed on OpenReview:** *https://openreview.net/forum?id=N69lSYWkMw*

## Abstract

Spatiotemporal point processes (STPPs) are probabilistic models for events occurring in continuous space and time. Real-world event data often exhibit intricate dependencies and heterogeneous dynamics. By incorporating modern deep learning techniques, STPPs can model these complexities more effectively than traditional approaches. Consequently, the fusion of neural methods with STPPs has become an active and rapidly evolving research area. In this review, we categorize existing approaches, unify key design choices, and explain the challenges of working with this data modality. We further highlight emerging trends and diverse application domains. Finally, we identify open challenges and gaps in the literature.

## 1 Introduction

Real-world events, such as urban crime incidents, epidemic spread, earthquakes, and environmental changes, can be represented as sequences of discrete events with both spatial and temporal components. Studying the spatiotemporal distribution of events and discovering the relationships among different types of events is an increasingly important area of research for understanding the dynamics and mechanisms of the occurrence of events. One such paradigm is the spatiotemporal point process (STPP) model, defined as a stochastic process that describes the spatial and temporal distribution of discrete events (Daley and Vere-Jones, 2007), which is well suited to capture the complex relationships between events, including self-excitation, and the interactions between events and spatial covariates across time and space.

Reviews on neural point processes have focused on modeling the temporal dynamics of events using neural networks (NNs) (Shchur et al., 2021; Lin et al., 2021; 2022). In contrast, reviews that address spatiotemporal

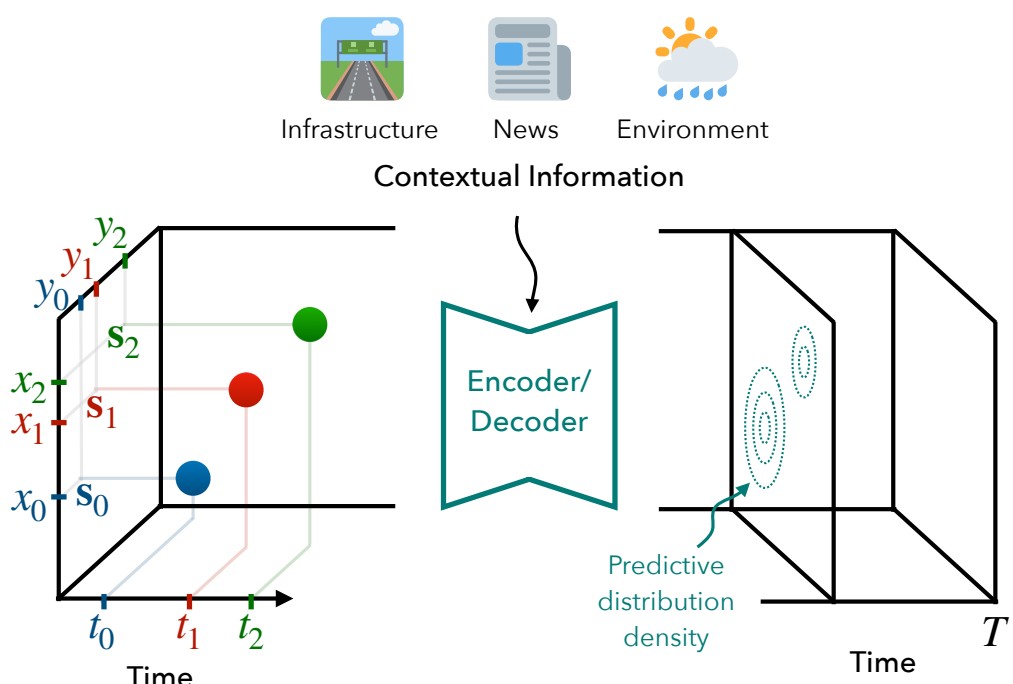

Figure 1: Schematic of the autoregressive construction of an STPP with two spatial dimensions. Three events are fed into an NN that predicts the likelihood of future event times and locations.

event modeling have focused on traditional statistical methods (González et al., 2016; Reinhart, 2018). More recently, some reviews have explored specific aspects of the use of machine learning in STPPs (Wikle and Zammit-Mangion, 2023; Bernabeu et al., 2024); these typically focus on specific aspects rather than providing comprehensive coverage. Meanwhile, work on neural STPPs has achieved remarkable progress.

Our work bridges this gap by systematically exploring design choices, methodological innovations, and key challenges in neural STPPs. To our knowledge, no prior survey has comprehensively examined these aspects in this context.

## 1.1 Why We Need Neural STPPs

Structural differences between space and time make modeling challenging. Time is unidirectional, while spatial propagation is omnidirectional and affected by environmental factors. Traditional methods rely on strong parametric assumptions and independence, limiting flexibility. They struggle with long-range dependencies, fail to capture heterogeneous dynamics across space and time, and cannot integrate multimodal data. Additionally, single-step predictions cause error accumulation. NN-based methods overcome these limitations by encoding space and time efficiently, handling dependencies, and learning heterogeneous patterns. They automate feature extraction, integrate diverse data, and scale effectively.

## 1.2 Scope and Structure of the Paper

This survey reviews neural STPPs, covering core models, applications, and key components in event modeling. We focus on studies using point processes with neural parameterization to capture spatiotemporal dynamics. Our literature search included keyword-based queries, citation tracking, and seminal works in neural temporal point processes. By outlining fundamental principles and design choices, we provide a practical foundation for researchers. We first introduce the necessary background and notation, then present available modeling choices, including architectures, training procedures, and metrics. The next section highlights notable applications of neural STPPs, and we conclude by discussing open challenges.

## 2 Background and Notation

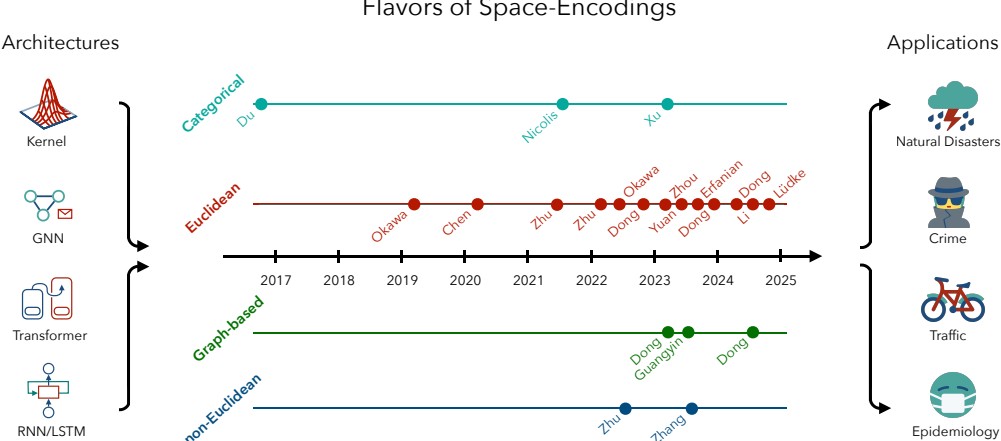

Figure 2: A timeline of the reviewed methodological and application-focused works, along with an overview of neural architectures and application domains. Papers are grouped based on how they encode spatial information, though in some cases these categorizations are not strictly defined.

### 2.1 Spatiotemporal Point Processes

STPPs are concerned with modeling sequences of random events in continuous space and time (Moller and Waagepetersen, 2003). A realization or sample of an STPP is defined up to a time horizon $T \in \mathbb{R}_{\geq 0}$. It is a finite, ordered event sequence containing pairs $X = [(t_1, \boldsymbol{s}_1), (t_2, \boldsymbol{s}_2), \ldots, (t_n, \boldsymbol{s}_n)]$, where $t_i \in [0, T]$ denotes the time and $\boldsymbol{s}_i \in \mathcal{S}$ the location of event $i$. Here, $\mathcal{S} \subseteq \mathbb{R}^d$ represents the spatial domain, which is typically a bounded region in $d$-dimensional Euclidean space, and typically $d = 2$. For a given event sequence, $\mathcal{H}_t = \{(t_i, \boldsymbol{s}_i) \mid t_i < t\}$ denotes the history of the current realization up to (but excluding) time point $t$ (cf. Figure 1).

### 2.2 Likelihood

An STPP can be specified by defining the likelihood of event sequences. The framework follows the temporal priority principle (Daley and Vere-Jones, 2007), which asserts that all causes must precede their effects. As a result, the likelihood of an event sequence $X$ can be expressed in an autoregressive form (Rasmussen, 2018):

$$f(X) = \underbrace{\left( \prod_{i=1}^{n} f^{\text{pred}}(t_i, \boldsymbol{s}_i \mid \mathcal{H}_{t_i}) \right)}_{\text{Likelihood of observed events}} \cdot \underbrace{\left( 1 - F^{\text{pred}}(T \mid \mathcal{H}_{t_n}) \right)}_{\text{Probability of no events after } t_n} \,,$$

where $f^{\text{pred}}(t, \boldsymbol{s} \mid \mathcal{H}_t)$ is the predictive distribution, specifying the conditional probability density function (PDF) for the next event occurring at a given timestamp $t$ and location $\boldsymbol{s}$, given the history of past events $\mathcal{H}_t$. The term $F^{\text{pred}}(T \mid \mathcal{H}_{t_n})$ represents the cumulative distribution function (CDF) of the predictive distribution, which gives the probability that an event occurs before or at time $T$, regardless of $\boldsymbol{s}$. Thus, $1 - F^{\text{pred}}(T \mid \mathcal{H}_{t_n})$ is the probability of no events occurring after the last observed event. Conveniently, this formulation also provides a principled approach for simulation (i.e., the generation of samples from the underlying stochastic model).

### 2.3 Intensity Function

In practice, an STPP is often described using a (conditional) intensity function (CIF) instead of a predictive distribution; they are mutually translatable (Chen et al., 2021). The CIF $\lambda(t, \boldsymbol{s} \mid \mathcal{H}_t)$, denoted by $\lambda^*(t, \boldsymbol{s})$ as a shorthand for its dependence on $\mathcal{H}_t$, is defined as:

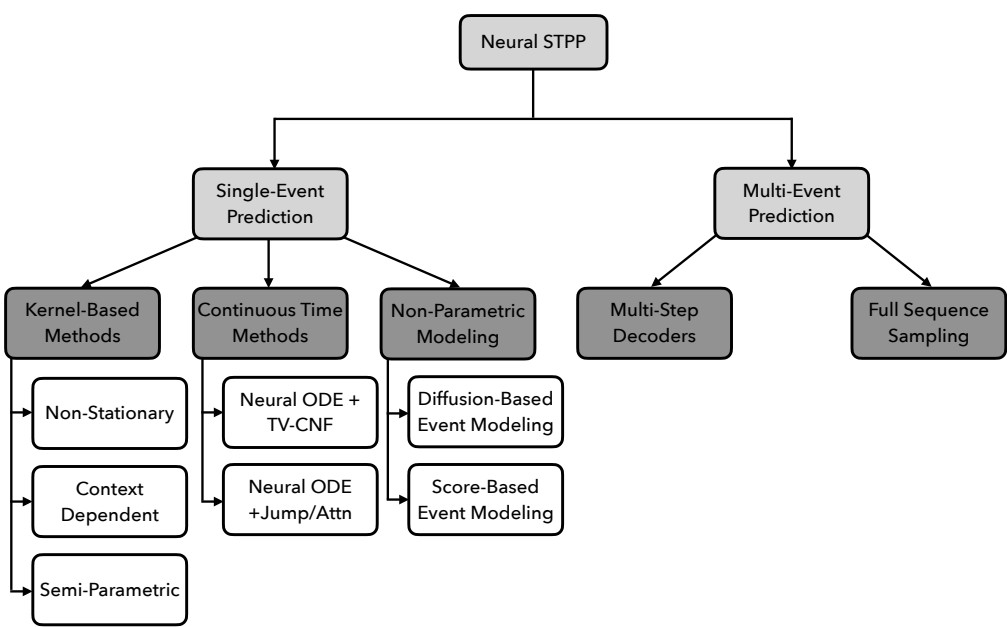

Figure 3: Taxonomy of neural STPP models organized by prediction task. Single-event prediction includes kernel-based and continuous-time and generative approaches, while multi-event prediction covers multi-step decoders and full-sequence or set-based models. History encoders (e.g., RNNs, Transformers) are not shown here, as they are architectural components that can be used across both categories.

$$\lambda^*(t, \boldsymbol{s}) = \lim_{\Delta_{\boldsymbol{s}}, \Delta_t \downarrow 0} \frac{\mathbb{P}\big(\text{Event} \in \big(B(\boldsymbol{s}, \Delta_{\boldsymbol{s}}) \times [t, t + \Delta_t)\big) \mid \mathcal{H}_t\big)}{|B(\boldsymbol{s}, \Delta_{\boldsymbol{s}})| \, \Delta t}, \tag{1}$$

where $B(\boldsymbol{s}, \Delta_s)$ is a $d$-dimensional ball (a disk if $d = 2$) centered at $\boldsymbol{s} \in \mathcal{S}$ with radius $\Delta_s$, and $|B(\boldsymbol{s}, \Delta_s)|$ denotes its volume.

Events can include additional information beyond location as a mark (Daley and Vere-Jones, 2007). While the core concept remains unchanged, this approach incorporates multimodal contextual data.

## 3 Modeling Neural Spatiotemporal Point Processes

Neural STPPs use neural networks to model event evolution and occurrence probabilities, thereby implicitly defining a probability density function over event sequences. The choice of modeling approach directly affects training efficiency, tractability of likelihood evaluation, and the quality of generated events. A common design pattern is to construct a latent representation of the event history and decode it to generate future events.

Spatial encoding strategies are a central challenge in STPPs, as they strongly influence model performance across all modeling paradigms. The representation of space determines how heterogeneity, context, and inter-event interactions are captured. Temporal encoding methods are not a primary focus here, as they are already well covered in prior surveys (Shchur et al., 2021).

Modeling choices are often driven by the application context. For example, crime forecasting may prioritize modeling the influence of specific landmarks, earthquake prediction may require fine-grained spatial accuracy, and epidemic modeling might emphasize regional policy effects, favoring coarser spatial resolution. Some applications impose parametric constraints to enable hypothesis testing, while safety-critical settings may require explicit uncertainty quantification. These considerations shape the choice of spatial encoding strategy.

Discretization-based methods simplify modeling but can limit the ability to encode inductive biases such as a stronger influence from nearby events. Their performance can also be sensitive to spatial granularity, boundary effects, and local correlations. Graph-based approaches, such as graph neural networks (GNNs), alleviate these issues by modeling interactions between discrete spatial cells.

Continuous-space methods, on the other hand, represent locations via raw coordinates (e.g., latitude–longitude) but risk ignoring anisotropic propagation patterns. To address this, models may employ location-specific parametric kernels, incorporate contextual covariates (e.g., satellite imagery or geotagged text), or learn spatial embeddings that capture relational structure. Attention mechanisms can implicitly model heterogeneous spatial effects, while GNNs and heterogeneous kernels provide explicit representations of spatial interactions.

**Section organization.** Our categorization in this section reflects both architectural and task-specific considerations, integrating the spatial modeling strategies discussed above. Section 3.1 covers history event encoders, presented as architectural components that can be paired with a variety of decoding strategies. These encoders, implemented via RNNs, Transformers, or other architectures, serve as reusable mechanisms for representing past event sequences and can be applied in both single-event and multi-event prediction settings.

In contrast, Sections 3.2 and 3.3 are organized by prediction task. For single-event prediction, we include only modeling paradigms that are specific to this task, namely, kernel-based approaches, continuous-time neural ODE/CNF models, and non-parametric generative models, which require sequential likelihood evaluation for each predicted event. For multi-event prediction, we include only decoders that are explicitly designed to generate multiple events or predict entire event sequences; these may use any history encoding scheme or latent modeling approach, but their decoding objective is sequence-level rather than event-by-event.

## 3.1 History Event Encoder

Spatiotemporal event modeling requires a CIF that combines temporal evolution and spatial dependencies (cf. Figure 1). A common approach to predicting an event $(t_i, \boldsymbol{s}_i)$ is to first encode each historical event $(t_j, \boldsymbol{s}_j) \in \mathcal{H}_{t_i}$ as an embedding $\boldsymbol{e}_j = [\omega(t_j); \sigma(\boldsymbol{s}_j)]$. Some architectures directly use raw time $t_j$ and space $\boldsymbol{s}_j$, while others transform them via $\omega(\cdot)$ (e.g., linear, trigonometric, or logarithmic mappings) and $\sigma(\cdot)$ (e.g., linear layers or one-hot encodings). A history encoder then processes embeddings into a latent state $\boldsymbol{h}_i$ to parameterize the CIF.

RNN variants like GRU and LSTM are common history encoders, updating states as $\boldsymbol{h}_{i+1} = \text{RNN}(e_i, \boldsymbol{h}_i)$ (Du et al., 2016; Omi et al., 2019; Shchur et al., 2020). Their sequential nature reduces storage needs but limits parallelization. Additionally, they suffer from gradient vanishing and long-term memory loss (Le and Zuidema, 2016).

Attention mechanisms (Vaswani et al., 2017) overcome several limitations of recurrent encoders. They have demonstrated superior performance as history encoders for temporal point processes (Zhang et al., 2020a; Zuo et al., 2020) and lately in STPPs (Zhou et al., 2022; Yuan et al., 2023). Nevertheless, the $\mathcal{O}(N^2)$ space complexity required to construct the attention matrix can pose practical challenges when dealing with very long event sequences.

## 3.2 Single Event Prediction

Neural STPPs predict the time and location of the next event. We review parametric, neural, mixture, diffusion, and continuous-time models, emphasizing those that capture spatial heterogeneity, contextual influences, and graph-based interactions.

As described in subsection 3.1, a common strategy is to learn a latent representation of the event history, often using recurrent networks, self-attention mechanisms, or continuous-time embeddings, and then directly predict the timing, location, or type of the next event from this latent state. Another approach is to parametrize the conditional intensity function, which describes the instantaneous rate of event occurrence given the history, and then use this function to sample or optimize for the most likely next event. A

third paradigm is non-parametric modeling of the intensity, which learns flexible influence functions without strong parametric assumptions, enabling the capture of complex dependencies while retaining interpretability. Finally, interaction-focused methods, such as kernel-based approaches, explicitly represent how past events of different types and positions shape the likelihood of future events.

Each paradigm offers different trade-offs. Direct latent-history prediction can be highly expressive and computationally efficient at inference time, but may sacrifice interpretability. Parametric intensity modeling preserves the probabilistic foundations of point processes and enables likelihood-based training, but often relies on restrictive assumptions. Non-parametric approaches relax these assumptions and flexibly capture complex dependencies, but typically require large datasets and offer limited interpretability. Interaction-focused methods, such as kernel-based models, provide clear influence functions and thus interpretability, but face scalability challenges in high-dimensional or dense event settings. Finally, continuous-time neural ODE or CNF models enable expressive spatiotemporal dynamics without discretization, but are computationally intensive due to repeated evaluations of differential equations.

### 3.2.1 Kernel-based Methods

Kernel-based methods fall under the interaction-focused paradigm. In this framework, past events influence the likelihood of future events through parameterized kernel functions that capture temporal, spatial, or spatiotemporal dependencies. This design offers several benefits: it provides a transparent mechanism for attributing predictions to specific past events, supports interpretability by making event-event relationships explicit, and can be extended to incorporate flexibility using deep learning to parameterize the kernels. Kernel-based methods are also well-suited to domains where pairwise triggering patterns are of interest, such as modeling aftershocks in seismology or contagion in epidemiology.

Formally, the conditional intensity is often defined as:

$$\lambda^*(t, \boldsymbol{s}) = \mu(t, \boldsymbol{s}) + \sum_{(t', \boldsymbol{s}') \in \mathcal{H}_t} K(t', t, \boldsymbol{s}', \boldsymbol{s}), \tag{2}$$

where $\mu(t, \boldsymbol{s})$ is the baseline rate, which can vary over time and space, and $K(\cdot)$ models the influence of past events. Traditional models use stationary kernels, assuming time- and space-invariant relationships (e.g., ETAS, Ogata, 1998; Hawkes, 1971), making strong parametric assumptions and missing heterogeneous effects.

Despite their strengths, kernel-based methods typically assume a specific functional form for the kernels (e.g., exponential, Gaussian), which can limit expressiveness and adaptability to complex data. They also tend to struggle with capturing long-range dependencies, as the influence of distant past events is often forced to decay to negligible values under standard kernel choices. Furthermore, performance may degrade in sparse data regimes, and the computational cost of evaluating interactions can grow quadratically with the number of historical events unless approximation techniques are used. Recent work on deep kernel learning and non-parametric kernel estimation aims to address some of these limitations by learning more flexible influence functions while retaining interpretability.

**Non-stationary kernels.** Designing the influence kernel $K(\cdot)$ in Equation 2 is crucial for capturing how past events trigger future occurrences. Neural parameterizations of $K(\cdot)$ enable non-stationary dependencies that vary across space, time, and contextual marks, moving beyond fixed parametric forms.

A representative approach is the Gaussian mixture model of Zhu et al. (2021b), where $K(\cdot)$ is expressed as:

$$K(t', t, \boldsymbol{s}', \boldsymbol{s}) = \sum_{l=1}^{L} \phi_{\boldsymbol{s}'}^{(l)} \cdot g(t, t', \boldsymbol{s}, \boldsymbol{s}' \mid \Sigma_{\boldsymbol{s}'}^{(l)}, \mu_{\boldsymbol{s}'}^{(l)}). \tag{3}$$

A neural network embeds spatial coordinates $\boldsymbol{s}$ to generate location-specific parameters $\mu_{\boldsymbol{s}}^{(l)}$, $\Sigma_{\boldsymbol{s}}^{(l)}$, and mixture weights $\phi_{\boldsymbol{s}}^{(l)}$ where $l \in 1, \ldots, L$ with $L$ denoting number of mixture components. Applying constraints ensures physical interpretability and reflects heterogeneous event diffusion. Visualizing learned kernels reveals region-specific influence propagation, making $K(\cdot)$ a smooth, adaptive function.

Alternate formulations consider heterogeneous interactions between events modeled using Mercer's theorem and neural basis functions (Zhu et al., 2022), leverage low-rank decomposition and a deep non-stationary influence kernel (Dong et al., 2023a), or embed events in graphs for non-Euclidean interactions (Dong et al., 2023b). These can be generalized as:

$$K(t', t, \boldsymbol{s}', \boldsymbol{s}) = \sum_{r=1}^{R} \sum_{l=1}^{L} \alpha_{rl} \psi_l(t', t) \phi_r(\boldsymbol{s}', \boldsymbol{s}), \tag{4}$$

where $\psi_l(\cdot)$ and $\phi_r(\cdot)$ are neural feature maps modeling time displacement, spatial relations, or graph connectivity, with $\alpha_{rl}$ scaling their contributions. $L$ and $R$ refer to the number of basis kernels used to decompose the kernel in Equation 2.

For instance, Dong et al. (2023c) and Dong and Xie (2024) use learned Gaussian bases for anisotropic geography, while Dong et al. (2024) combine a stationary Gaussian kernel with a GNN mark kernel to capture network constraints and landmark effects.

**Context.** Event dynamics are influenced by contextual covariates like georeferenced images or text. Okawa et al. (2019) define event intensity as a spatially localized mixture of kernels:

$$\lambda^*(t, \boldsymbol{s} \mid \mathcal{D}) = \sum_{j=1}^{M} f(\boldsymbol{u}_j, \boldsymbol{z}_j; \theta) K(t, \boldsymbol{s}, \boldsymbol{u}_j), \tag{5}$$

where $K(\cdot)$ is a compactly supported Gaussian kernel, $\mathcal{D}$ is a set of contextual features, and $\boldsymbol{u}_j$ are spatiotemporal anchors uniformly distributed in time and space. Contextual features $\boldsymbol{z}_j$, extracted from $\boldsymbol{u}_j$, inform mixture weights $f(\boldsymbol{u}_j, \boldsymbol{z}_j; \theta)$ through a deep network combining image and text embeddings. These weights adapt to heterogeneous conditions (e.g., urban infrastructure, social events), enabling dynamic kernel weighting centered on $u_j$ while restricting contextual influence to local neighborhoods.

Zhang et al. (2023) replace the parametric kernel with a deep kernel $K(t, \boldsymbol{s}, \boldsymbol{u}) = k_\phi(g(t, \boldsymbol{s}), g(\boldsymbol{u}))$, where $g(\cdot)$ is a non-linear transformation by a deep NN, learning complex spatial correlations beyond standard Euclidean distance.

Okawa et al. (2022) propose a method for incorporating high-dimensional contextual data into the Hawkes process. They introduce a weighting term in the excitation kernel, extracting relevant features using CNNs and employing continuous kernel convolution to transform discretized features into continuous space. This enables capturing spatial heterogeneity and external influences while ensuring tractable optimization.

**Semi-parametric kernels.** Zhou et al. (2022) introduce a non-parametric mixture-based intensity given by

$$\lambda^*(\boldsymbol{s}, t \mid \boldsymbol{z}) = \sum_{i=1}^{n+J} w_i \, K_{\boldsymbol{s}}(\boldsymbol{s}, \boldsymbol{s}_i; \gamma_i) \, K_t(t, t_i; \beta_i), \tag{6}$$

where each kernel $K_{\boldsymbol{s}}(\cdot)$ and $K_t(\cdot)$ is a normalized radial basis function, and the parameters $\{w_i, \gamma_i, \beta_i\}$ are drawn from a latent process $\boldsymbol{z}$. The history is encoded by a Transformer (subsection 3.1) and decoded to parameterize the latent process. The samples of the latent process are further decoded via a feedforward network. This method captures uncertainty in the timing and spatial location of events. By augmenting the observed $n$ events with $J$ randomly sampled representative points, this approach addresses global background intensity, thereby reducing the reliance on strong parametric assumptions.

### 3.2.2 Continuous Time-Based Methods

While the previous section examined methods that explicitly capture event–event relationships, we now turn to continuous-time approaches that directly model the conditional intensity function. These methods enable inference over event distributions in space and event rates at arbitrary time points. Kernel density estimators (KDEs) have traditionally been used to model the spatial distribution of events. While simple to implement,

this approach often fails to capture the complex and history-dependent nature of spatial patterns observed in real-world spatiotemporal processes. In parallel, kernel-based interaction models (subsubsection 3.2.1; such as those used in Hawkes processes) focus on explicitly modeling the triggering influence between events through predefined spatial and temporal kernels, which specify how past events affect the likelihood of future ones.

Continuous time-based CNF models take a different route: rather than summing kernels or explicitly parameterizing triggering, they directly model the spatial distribution of events and how it evolves over continuous time. Following Chen et al. (2021), the conditional intensity is factorized into temporal and spatial terms,

$$\lambda^*(t, x) = \lambda^*(t)\, p^*(x \mid t), \tag{7}$$

where $\lambda^*(t)$ captures the overall event rate in time and $p^*(x \mid t)$ captures where in space events occur at time $t$. This decomposition models space and time independently, potentially missing interaction effects. However, this is addressed by making $p^*(x \mid t)$ also depend on previous events and time in extension to the basic modeling technique.

**Hidden-state dynamics and ground intensity.** To model history-dependence, a continuous-time hidden state $h_t$ summarizes all past events. This hidden state evolves smoothly between events and jumps instantaneously when new events occur:

$$h_{t_0} = h_0, \qquad \frac{dh_t}{dt} = f_h(t, h_t) \quad \text{for } t \neq t_i, \tag{8}$$

$$\lim_{\varepsilon \to 0} h_{t_i + \varepsilon} = g_h\big(t_i, h_{t_i}, x_{t_i}\big), \tag{9}$$

$$\lambda^*(t) = g_\lambda(h_t) \quad \text{(e.g., softplus head to enforce positivity).} \tag{10}$$

Equation 8 defines smooth dynamics via a neural ODE; Equation 9 injects discrete updates at event times; and Equation 10 maps $h_t$ to the ground intensity. This lets the temporal process condition on past events in a continuous-time fashion.

**Static (time-independent) CNF for space.** A standard CNF learns a static spatial density, the same for all $t$, by transporting a base density $p(x_0)$ through an artificial flow time $s$:

$$\frac{dx_s}{ds} = f(x_s), \tag{11}$$

$$\log p(x_s) = \log p(x_0) - \int_0^s \text{tr}\left(\frac{\partial f}{\partial x}(x_\sigma)\right) d\sigma. \tag{12}$$

Here, $f$ is a neural network that acts like a "velocity field" describing how each point in space should move during the transformation. Equation 12 uses the instantaneous change-of-variables formula to track how the density changes along the flow. Because $f$ does not depend on $t$, the learned distribution $p(x)$ is stationary over real time.

**Time-Varying CNF (TV-CNF).** The TV-CNF relaxes the stationarity assumption by making the velocity field explicitly depend on $t$:

$$\frac{dx_t}{dt} = f(t, x_t), \tag{13}$$

$$\log p^*(x_t \mid t) = \log p(x_0) - \int_0^t \text{tr}\left(\frac{\partial f}{\partial x}(\tau, x_\tau)\right) d\tau. \tag{14}$$

Now the spatial distribution can smoothly change with real time, enabling the modeling of non-stationary phenomena (e.g., a hotspot that drifts across space). Because the model ignores history, trajectories for multiple events can be computed in parallel using the reparameterization:

$$\frac{d}{ds}\begin{bmatrix} x_s^{(1)} \\ \vdots \\ x_s^{(n)} \end{bmatrix} = \begin{bmatrix} t_1\, f(st_1, x_s^{(1)}) \\ \vdots \\ t_n\, f(st_n, x_s^{(n)}) \end{bmatrix}, \tag{15}$$

which integrates all event locations $\{x_{t_i}^{(i)}\}$ in a single ODE solve.

**Jump CNF (history-conditioned space).** Jump CNFs incorporate history into the spatial model by conditioning on $h_t$ and allowing instantaneous jumps in location space at event times:

$$x_0 \sim p(x_0), \qquad \frac{dx_t}{dt} = f_x(t, x_t, h_t) \quad \text{for } t \neq t_i, \tag{16}$$

$$\lim_{\varepsilon \to 0} x_{t_i + \varepsilon} = g_x(t_i, x_{t_i}, h_{t_i}). \tag{17}$$

The log-likelihood now has continuous-flow terms and discrete jump terms:

$$\log p^*(x_t \mid t) = \log p(x_0) + \sum_{i=1}^{n} \left( -\int_{t_{i-1}}^{t_i} \text{tr}\left( \frac{\partial f_x}{\partial x}(\tau, x_\tau, h_\tau) \right) d\tau - \log \left| \det \frac{\partial g_x}{\partial x}(t_i, x_{t_i}, h_{t_i}) \right| \right)$$

$$- \int_{t_n}^{t} \text{tr}\left( \frac{\partial f_x}{\partial x}(\tau, x_\tau, h_\tau) \right) d\tau. \tag{18}$$

The second term in the equation models the change in density up to the last event, and the last term models the change in density from the last event. The integrals account for gradual deformation between events, and the log-determinants capture instantaneous changes at jumps. Because jumps require restarting the ODE solver, computation scales linearly with the number of past events.

**Attentive CNF (parallel, history-conditioned space).** The Attentive CNF uses attention to couple all event trajectories and their histories inside a single parallel ODE system:

$$\frac{d}{ds} \begin{bmatrix} x_s^{(1)} \\ \vdots \\ x_s^{(n)} \end{bmatrix} = f_{\text{Attn}}(s, X_s, H), \tag{19}$$

where $f_{\text{Attn}}$ is a Lipschitz multihead-attention network over past events and $H$ denotes their hidden states. This retains history-dependence but avoids the sequential cost of Jump CNFs by evolving all trajectories together.

In contrast to kernel-based methods (subsubsection 3.2.1), which rely on explicitly defined influence functions, CNF-based approaches adopt a generative viewpoint by parameterizing the continuous evolution of event distributions through neural differential equations. Rather than prescribing parametric kernels, these models learn transformations from simple base distributions into complex spatiotemporal event patterns. The conditional intensity function is then derived from the latent trajectory defined by the flow, enabling highly flexible modeling of non-stationary and heterogeneous event dynamics.

The key advantage of CNFs lies in their expressiveness: they capture intricate dependencies in continuous time without the need for handcrafted kernel forms. This flexibility makes them particularly suitable for domains where interactions are irregular, long-range, or anisotropic. At the same time, CNFs unify density estimation and generative modeling, providing a principled framework for both likelihood-based inference and sample generation.

However, this expressiveness comes with trade-offs. Training CNF-based STPPs typically involves solving ordinary differential equations with neural network dynamics, which introduces significant computational overhead compared to kernel methods. Likelihood estimation requires backpropagating through ODE solvers and computing trace terms, which can suffer from numerical instability. Moreover, while kernel-based approaches yield interpretable influence functions, the latent representations learned by CNFs are implicit, making it harder to attribute event dependencies or quantify influence in an interpretable manner.

### 3.2.3 Non-parametric modeling

Neural STPP models often rely on restrictive assumptions, such as conditional independence or unilateral dependence between the distributions of temporal and spatial events. These assumptions limit their ability to accurately predict events in real-world scenarios, where events exhibit complex interdependencies in both time and space.

**Diffusion-based event modeling.** To overcome the limitations of conditional independence or unilateral dependence in neural STPPs, Yuan et al. (2023) introduce Spatiotemporal Diffusion Point Processes (DSTPP), which directly learn the joint distribution of temporal and spatial events via a denoising diffusion probabilistic model (Ho et al., 2020) without structural constraints. A spatiotemporal self-attention encoder separately embeds time and space, fuses them into history representations, and conditions the diffusion process. The forward process gradually perturbs event coordinates with Gaussian noise, while the reverse process iteratively denoises them using a co-attention network that dynamically captures mutual spatiotemporal dependencies, enabling joint distribution learning without assuming independence or requiring integrable intensity functions. At each reverse step $k$, the model predicts

$$p_\theta\left(x_i^{(k-1)} \mid x_i^{(k)}, h_{i-1}\right) = p_\theta\left(\tau_i^{(k-1)} \mid \tau_i^{(k)}, s_i^{(k)}, h_{i-1}\right) \ p_\theta\left(s_i^{(k-1)} \mid \tau_i^{(k)}, s_i^{(k)}, h_{i-1}\right), \tag{20}$$

where $x_i^{(k)}$ denotes the noisy state at step $k$. Model parameters $\theta$ are learned by minimizing the denoising objective

$$\mathcal{L} = \mathbb{E}_{x_i^0, \epsilon, k}\left[\left\|\epsilon - \epsilon_\theta\left(\sqrt{\alpha_k}\, x_i^0 + \sqrt{1-\alpha_k}\, \epsilon, \ h_{i-1}, k\right)\right\|^2\right], \tag{21}$$

where $x_i^0$ is the clean event, $\epsilon \sim \mathcal{N}(0, I)$ is Gaussian noise, $\alpha_k = 1 - \beta_k$ is the variance schedule, and $\epsilon_\theta(\cdot)$ is the learned denoising network. For spatial decoding, the model directly predicts continuous coordinates or can apply a rounding step for discrete locations. This method eliminates the need for approximation during sampling and supports continuous and discrete spatial domains.

**Score-based spatiotemporal modeling.** While diffusion-based STPPs avoid restrictive assumptions, they require multi-step denoising with carefully tuned noise schedules. To provide a simpler yet expressive alternative, Li et al. (2024) introduce SMASH, which directly models future events via their conditional score functions. Instead of parameterizing intensities or perturbing trajectories with diffusion noise, SMASH represents the temporal distribution through the score $\psi_t^i(t \mid k) = \partial_t \log p^i(t \mid k)$ and the spatial distribution through $\psi_x^i(x \mid t, k) = \nabla_x \log p^i(x \mid t, k)$. Marks are modeled separately with a categorical distribution $p_i(k \mid t) \propto \lambda_i(t, k)$. This decomposition allows the model to capture complex dependencies across time, space, and marks without assuming conditional independence or integrability of intensities.

New events are generated via Langevin dynamics,

$$x^{(n+1)} = x^{(n)} + \tfrac{\epsilon}{2}\, \psi(x^{(n)}) + \sqrt{\epsilon}\, z_n, \tag{22}$$

where $\psi(\cdot)$ is the relevant score function, $\epsilon$ is the step size, and $z_n \sim \mathcal{N}(0, I)$ is Gaussian noise. This iterative refinement moves noisy samples toward regions of high model density, yielding continuous event times and locations as well as calibrated mark probabilities. As a result, SMASH provides not only point predictions but also principled uncertainty quantification, producing confidence intervals for event times and confidence regions for locations. Compared to Yuan et al. (2023), who model perturbed distributions via denoising diffusion with multiple noise scales, SMASH achieves comparable flexibility through direct score-based modeling, offering a more efficient and uncertainty-aware generative framework.

Despite their flexibility, diffusion- and score-based approaches to STPPs have notable drawbacks compared to more classical kernel-based and continuous-time formulations. First, both diffusion and score matching rely on iterative sampling procedures (multi-step denoising or Langevin dynamics), which are substantially more expensive than the closed-form sampling available in kernel-based models such as Hawkes processes. This limits their scalability when simulating long event sequences. Second, unlike continuous-time intensity models, which provide explicit hazard rates and interpretable dynamics, score-based generative models treat event prediction as implicit density estimation. While this allows them to capture complex dependencies, it also makes interpretability and direct likelihood evaluation more difficult. Finally, these methods require careful tuning of noise schedules (diffusion) or step sizes (Langevin dynamics), and their performance may degrade under distributional shifts where kernel-based methods retain robustness through their parametric structure. As a result, diffusion- and score-based models trade analytical tractability and efficiency for greater modeling flexibility and uncertainty quantification.

### 3.3 Multi-Event Prediction

Single-event prediction methods estimate the CIF one step at a time. However, predicting multiple future events becomes computationally expensive because it requires repeated integration over time and space. This process also involves updating the history after each prediction, which can lead to errors that accumulate over longer sequences. Two recent approaches tackle these problems from different angles. The first approach employs multi-step decoders that predict several future events simultaneously, reducing both computational overhead and the risk of error accumulation. The second approach bypasses the CIF entirely by modeling entire sequences through diffusion, which allows for parallel sampling and flexible conditioning on partial observations.

#### 3.3.1 Multi-Step Decoders

Rather than predicting one event at a time, Erfanian et al. (2022) introduce a Transformer-based model that generates the next $L$ events in a single forward pass. The model takes as input a history of $n$ past events, each with a timestamp and spatial location, and encodes this into hidden representations that capture spatiotemporal dependencies. The decoder then produces probability distributions for the next $L$ events. At each future step $l \in [n+1, \ldots, n+L]$, it predicts both the inter-event time $t_l$ given the history and the location $x_l$ conditioned on the history and the sampled $t_l$. Time is modeled with an exponential base distribution and space with a multivariate Gaussian, both made more flexible through normalizing flows: a softsign bijector for time and a RealNVP flow (Dinh et al., 2017) for space. The outputs are full probability distributions over when and where events may occur. Predicting them in parallel avoids the repeated integrations of classical point processes and reduces the step-by-step error accumulation of autoregressive rollouts.

Multi-event forecasting approaches of this kind illustrate the trade-offs of moving beyond single-step prediction. Parallel decoding offers efficiency and reduces long-horizon drift, but also raises challenges. Each step has its own probabilistic head, conditioned on shared hidden states, which allows interactions to be modeled implicitly but weakens direct coupling across horizons. This can lead to less coherent trajectories when rolled out generatively. Performance also depends on how well the base distributions align with the data, and mismatches must be absorbed by the flows, which can reduce stability. Evaluation adds further complexity. Erfanian et al. (2022) for example, condition spatial likelihoods on the true future times during testing, which provides stronger supervision than would be available in a fully generative rollout. Finally, multi-event predictors often show higher sensitivity to hyperparameters and regularization than single-step models, limiting robustness across datasets.

Single-step models, by contrast, are straightforward and often easier to interpret, since each prediction is conditioned directly on the observed past. They tend to perform reliably at short horizons but suffer from error accumulation when rolled out over many steps. Multi-event models mitigate this compounding of errors and improve efficiency, but their joint forecasts are less transparent and sometimes harder to interpret. The choice between them is therefore a balance between short-term stability and interpretability on one side, and longer-range efficiency on the other.

#### 3.3.2 Sampling Full Sequences

A different line of work models entire event sequences rather than predicting them one at a time. Lüdke et al. (2023) introduce a diffusion-based framework where the basic modeling unit is the set of events. Let a sequence be represented as a set $S = \{(t_i, x_i)\}_{i=1}^{n}$, where $t_i$ denotes the timestamp and $x_i$ the spatial location (or a more general element in a metric space). The forward process gradually corrupts this set by perturbing event coordinates and randomly removing points, yielding a sequence of noisy sets $\{S_t\}_{t=0}^{T}$. The reverse model is then trained to reconstruct $S_0$ from $S_T$ by approximating the conditional distributions $p_\theta(S_{t-1} \mid S_t)$, $t = T, \ldots, 1$.

This formulation parametrizes events collectively rather than individually. Instead of specifying an intensity function $\lambda(t, x \mid H_t)$ or predicting the next event $(t_{n+1}, x_{n+1})$, the model learns a transformation that maps noisy sets back to realistic point patterns. In practice, events are represented as continuous densities, for example, by approximating each point mass $\delta_{(t_i, x_i)}$ with a Gaussian kernel. The diffusion dynamics

Table 1: Parameter estimation methods for Neural Spatiotemporal Point Processes

| Method | Description | Strengths | Limitations / Trade-offs |
|---|---|---|---|
| **Maximum Likelihood Estimation (MLE)** | Fits parameters by maximizing likelihood (or minimizing NLL); requires evaluating the CIF and its integral over space-time. | Statistically efficient; well-understood; direct probabilistic interpretation; widely used baseline. | Integral often intractable; needs numerical/MC approximations; expensive in high-dimensional STPPs; sensitive to misspecification. |
| **Amortized Variational Inference (AVI)** | Learns an approximate posterior by maximizing an ELBO; often paired with kernel-based intensities for tractable integration. | Scales to irregular continuous-time data; supports latent variables; sometimes admits closed-form integrals. | Quality depends on the variational family; risk of posterior collapse, and added modeling/implementation complexity. |
| **Score-Matching (SM)** | Matches gradients of log-density (score) between model and data; DSM variant adds noise for stability and avoids second derivatives. | Avoids explicit normalization and integrals; flexible, non-parametric; naturally supports uncertainty quantification. | Requires differentiability; may be less sample-efficient than MLE; DSM noise requires careful tuning. |
| **Score-Matching Pseudolikelihood** | Decomposes the joint into conditionals; applies score matching for times/locations and conditional likelihood for marks. | Handles intractable normalizers; mitigates over/under-confidence via calibrated posteriors; supports marked STPPs. | Needs iterative sampling (e.g., Langevin); computationally heavy at scale; sensitive to the chosen decomposition. |
| **Reinforcement / Imitation Learning** | Trains a generative policy to match data sequences using a distance (e.g., MMD) instead of likelihood. | No likelihood/integral computation; robust to model mismatch; can optimize task-specific rewards. | Requires well-chosen distance/reward; may underuse probabilistic structure; potential instability without careful tuning. |

then operate directly on these densities, allowing the model to capture both temporal and spatial structure in a unified way. Lüdke et al. (2025) extend this formulation to general metric spaces by decomposing the dynamics into thinning (removing points) and superposition (adding points), which provides a more interpretable view of how events evolve through the diffusion process. Conditioning is implemented with binary masks, making it straightforward to restrict sampling to specific windows or regions without retraining the model.

This parameterization offers notable benefits: events are modeled jointly as a set, long-range dependencies are naturally captured, and sampling is parallel rather than sequential. However, there are also drawbacks. First, the representation of point masses via smooth kernels introduces an approximation that may limit precision, particularly in high-resolution or high-dimensional settings. Second, because the model acts on the entire set, interpretability is reduced: the influence of specific past events on future outcomes is less transparent than in single-step or intensity-based models. Finally, while sampling time scales more favorably than autoregressive rollouts, the need for multiple reverse steps still makes the procedure less efficient in practice than direct one-step prediction.

Compared to single-step models, which explicitly predict $(t_{n+1}, x_{n+1})$ conditioned on the observed history $H_n$, diffusion models trade local precision and interpretability for the ability to model entire futures jointly. They often provide more coherent long-range forecasts, but at the cost of less transparency and weaker short-term accuracy.

## 4 Parameter Estimation and Inference

The key objectives of event prediction include enhancing predictive performance, improving generalization and robustness, understanding event dynamics through learned parameters, accounting for event behavior

heterogeneity, and capturing the influence of external factors. Predicting future events from a learned model requires sampling from its intensity function. Traditional statistical methods often rely on strong parametric assumptions for modeling event intensities, using techniques such as likelihood-based methods, partial likelihood, the EM algorithm, or Bayesian approaches.

In statistical inference, **Maximum Likelihood Estimation** (MLE) is most commonly used to fit classical and neural STPPs, typically by maximizing the likelihood function or, equivalently, minimizing the negative log-likelihood (NLL). For an observed sequence of $N$ events, the NLL is given by (Daley et al., 2003):

$$
\mathcal{L}_{\text{NLL}} = - \underbrace{\sum_{i=1}^{N} \log \lambda^*(t_i, \boldsymbol{s}_i)}_{\text{Log-likelihood of observed events}} + \underbrace{\int_{[0,T]} \int_{\mathcal{S}} \lambda^*(\tau, \boldsymbol{u}) \, d\tau \, d\boldsymbol{u}}_{\text{Expected number of events}}.
$$

When using neural networks to parameterize the CIF, evaluating the integral term is typically intractable. This complexity often requires numerical methods (Chen et al., 2021) or Monte Carlo methods (Mei and Eisner, 2017) for likelihood evaluation. However, these strategies can be computationally expensive and prone to numerical errors, particularly in high-dimensional spatiotemporal domains. While certain simplifying assumptions, such as exponential decay (Du et al., 2016) and linear interpolation (Zuo et al., 2020), can lead to closed-form solutions or faster approximations, they often restrict the expressiveness of the model.

In the purely temporal setting, Zhou and Yu (2023) introduce a paradigm for efficient and non-parametric inference of TPPs. They approximate the influence function (cf. Equation 2) via a NN, using **automatic integration** to compute its integral. A monotonically increasing integral network is trained; its partial derivative defines the CIF. This approach directly yields the CIF and its antiderivative from the network parameters, avoiding functional form restrictions. Building on this foundation, Zhou and Yu (2024) address the computational challenge of integrating the intensity function in 3D spatiotemporal domains by employing automatic integration. This approach learns an integral network, whose partial derivatives with respect to spatial and temporal inputs yield the intensity, ensuring an exact antiderivative without restricting the model's functional form. Furthermore, a ProdNet factorization of the influence function into 1D components enforces non-negativity while capturing spatiotemporal interactions. Maximizing the log-likelihood of observed events learns a highly expressive CIF, improving spatiotemporal event prediction.

Zhou et al. (2022) integrate flexible non-parametric modeling with **amortized variational inference**. This model captures events in continuous time, capturing irregular sampling dynamics and unifying spatial and temporal dependencies. By employing a kernel-based intensity function, the approach allows for closed-form integration, addressing previously intractable likelihood computations. This design avoids computationally expensive numerical integration inherent in neural ODE-based approaches (Chen et al., 2021). Furthermore, this non-parametric approach avoids restrictive parametric assumptions. Training via amortized variational inference maximizes the evidence lower bound (ELBO) of the likelihood while balancing reconstruction accuracy and posterior regularization. The framework uses the kernel-based intensity for gradient computation, facilitating end-to-end optimization of both encoder and decoder parameters.

Zhang et al. (2023) use **score matching**, which minimizes the Fisher divergence between the model's log-density gradient and the data's log-density gradient, thus bypassing the need to calculate the intractable integral. A denoising score matching (DSM) method is used, which improves stability by introducing a small amount of noise to the data, which avoids the computation of second derivatives. In contrast, Li et al. (2024) utilize a score-matching-based pseudolikelihood objective, which eliminates the need for explicit calculation of the normalizing term that makes the likelihood integral intractable. The model decomposes the joint intensity function into a product of conditional distributions, allowing for the application of score-matching techniques for event times and locations, while using a conditional likelihood for event marks. This approach is designed to overcome overconfidence and underconfidence by learning a posterior distribution that matches the actual data distribution, which also requires score-based sampling with Langevin dynamics.

**Reinforcement learning** (RL) frameworks provide a training approach that does not rely on likelihood calculations. Zhu et al. (2021b) employ an imitation learning framework to train their model. The learner

policy is defined by a PDF associated with the CIF of the point process and parameterized by the model's parameters. The goal is for this learner policy to replicate the expert policy reflected in the training data. The training process maximizes the expected reward, determined by the Maximum Mean Discrepancy (MMD) between empirical distributions of the training data and data generated by the learner's policy. This data-driven MMD reward function offers robustness to model mismatch as it compares data distributions instead of relying on a predefined likelihood. Additionally, the closed-form representation of the reward function enables computationally efficient optimization via analytical gradient calculation, avoiding intensive inverse reinforcement learning.

Each inference approach presents distinct trade-offs between computational cost, flexibility, and interpretability. A concise comparison of these methods, highlighting their respective advantages, limitations is provided in Table 1.

## 5 Evaluation Metrics

Evaluating STPPs requires task-specific metrics, summarised in Table 2. Model fit is often assessed using **NLL**, which measures how well predicted distributions align with observed data. However, NLL prioritizes overall distributional fit over individual event accuracy and can be biased due to computational approximations like Monte Carlo integration. Moreover, since NLL conditions on ground truth history, it is limited in evaluating true generative capacity.

Alternative distributional metrics address these shortcomings. **HD** (Zhou et al., 2022) directly compares learned and true distributions but requires ground truth intensity and spatial discretization, limiting real-world use, but useful for testing the ability of models to recover known patterns. **MMD** (Zhu et al., 2021b) avoids strong distributional assumptions by comparing generated and observed sequences but is computationally expensive and sensitive to kernel selection. MMD can also be used with **CD** (Lüdke et al., 2025) as the distance measure. This improves distributional comparisons and is useful for evaluating the generative performance of STPPs. **SL**, measured via Wasserstein distance (WD) between two categorical distributions, provides additional insight into model performance by comparing event sequence lengths, especially useful when the task involves measuring case counts, such as in epidemiology or crime modeling.

For point prediction accuracy, **MAE**, **MSE**, and **RMSE** are commonly used. MAE is robust to uniform errors, while RMSE is more sensitive to large deviations. Aggregated event predictions rely on normalized MAE (NMAE) (Okawa et al., 2022) and **MAPE** (Okawa et al., 2019). While NMAE enables cross-scale comparisons, it depends on predefined spatiotemporal regions. MAPE, though intuitive, can be unstable near zero values. Mean relative error (MRE) (Dong et al., 2023a) assesses intensity differences but also suffers from instability near zero. Prediction accuracy (ACC) measures the correctness of event counts but does not capture spatial or temporal precision.

Uncertainty quantification is crucial for robust evaluation. **CS** and **ECE** (Li et al., 2024) assess how well predicted confidence intervals align with observed distributions, though ECE's binning requires adaptive methods for reliability.

Selecting metrics depends on the task and dataset. A comprehensive evaluation combines prediction accuracy, distributional fit, and uncertainty quantification.

While NLL has been the standard objective for training and evaluating STPPs, it has notable shortcomings (Shchur et al., 2021) when used as the sole evaluation criterion. First, computing the NLL requires tractable access to the conditional intensity function and its integral. For many neural STPPs, this integral is intractable and must be approximated through Monte Carlo sampling, numerical quadrature, or variational bounds, which complicates reproducibility and makes comparisons across models less reliable. Second, NLL primarily reflects how well a model predicts individual events conditioned on past observations, but it does not necessarily capture how faithfully the model reproduces the joint distribution of sequences or higher-order spatio-temporal dependencies.

To address these limitations, recent work increasingly adopts sample-based distributional metrics such as the Wasserstein distance (WD) and MMD. Unlike NLL, these metrics operate directly on sets of generated

Table 2: Common Evaluation Metrics for Neural Spatiotemporal Point Processes

| Name | Definition | Advantages |
|------|-----------|-----------|
| **NLL** | Negative Log-Likelihood (likelihood of observed data given the predicted distribution). | Directly tied to MLE-based training; widely adopted for distributional fit. |
| **HD** | Hellinger Distance (a measure of distance between two probability distributions). | Fine-grained assessment of similarity; often used with known ground-truth distributions. |
| **MMD** | Maximum Mean Discrepancy (kernel-based comparison of two sample sets). | Distribution-agnostic; captures higher-order differences in generative quality. |
| **MAE / MSE / RMSE** | Mean/Absolute/Squared/Root Errors for point predictions (time or space). | Straightforward and interpretable; highlight large errors (MSE/RMSE) or typical errors (MAE). |
| **MAPE** | Mean Absolute Percentage Error (ratio-based prediction error). | Scale-independent; intuitive for relative errors across different magnitudes. |
| **SL** | Sequence Length (Wasserstein Distance between the categorical distribution of event sequence lengths). | Suitable for applications where the event count within an interval outweighs spatial and temporal accuracy. |
| **CD** | Counting Distance (A generalization of Wasserstein distance for order TPPs to STPPs using $L^1$ distance) | Useful for evaluating the generative performance of a model, especially for multi-step predictions |
| **CS / ECE** | Calibration Score / Expected Calibration Error (comparison of predicted vs. observed confidence). | Evaluates how well probability estimates reflect true event frequencies (calibration). |

and observed sequences, providing a more faithful assessment of generative quality. WD leverages the underlying geometry of space and time, making it well-suited to evaluating event distributions with structured dependencies, while MMD provides a flexible kernel-based test that is computationally efficient and model-agnostic. Together, these metrics complement NLL by shifting the focus from pointwise predictive fit toward the overall realism and fidelity of generated event sequences.

That said, NLL remains valuable in contexts where likelihood-based inference is required, such as hypothesis testing, model comparison under shared parametric assumptions, or applications where interpretability of the intensity function is central. Thus, a balanced evaluation strategy should combine NLL with distributional metrics like WD and MMD to provide a fuller picture of model performance.

# 6 Applications

The existing literature on neural STPPs mainly underscores their applications in settings where forecasts of when and where events occur support operational decisions. We organize this section by domain: natural disasters, crime, traffic, and epidemiology, because each area requires different modeling choices concerning space, time, and contextual factors. In practice, many models combine a background rate that carries covariates or marks with a self–exciting term that encodes event history. Spatial structure is represented either in continuous coordinates or along road and street networks, and temporal dynamics are modeled in continuous time. Contextual information, such as environmental conditions, demographic attributes, traffic states, or policy interventions, enters through the background or the marks.

## 6.1 Natural Disasters

**Earthquakes.** Nicolis et al. (2021) study daily forecasting for Chile, aiming to predict the maximum seismic rate for the following day and the macro-zone in which this maximum occurs. Their approach begins by estimating daily ETAS intensity maps on a regular grid. It then employs a recurrent network to predict the seismic rate based on recent ETAS summaries and a convolutional network to classify the macro-zone using recent ETAS images. Performance is reported using $R^2$ for rate prediction, accuracy for

zone classification, and error metrics such as MAE, MSE, and RMSE; the neural components are compared against ETAS using these measures. Zhang et al. (2024) propose CL-ETAS, a model that integrates ETAS forecasts within a ConvLSTM framework during both training and inference. They compare it with ETAS and ConvLSTM. They evaluate using the CSEP testing suite (number, magnitude, spatial, and likelihood tests) that is standard in earthquake forecasting, and report improvements under these tests relative to the baselines.

**Wildfires.** Xu et al. (2023) develop a marked, mutually exciting spatiotemporal Hawkes model designed for assessing binary ignition risk. The marks combine multimodal covariates, including static and dynamic environmental information, with both discrete and continuous features. Model parameters are estimated by an alternating optimization scheme, and evaluation is carried out on a large California dataset. Because ignitions are rare, they summarize detection with the F1 score. Additionally, they report conformal prediction sets for fire magnitude.

## 6.2 Crime

Dong and Xie (2024) model gunshot incidents in Atlanta with a non-stationary spatiotemporal Hawkes process. The temporal triggering is kept simple, while the spatial kernel is allowed to vary over the city and is represented through a neural construction that learns location–dependent shapes. Exogenous factors are represented by a location-dependent background intensity based on census covariates, leading to a conditional intensity that combines contagion and covariate effects. Parameters are estimated by MLE with gradient–based optimization, together with tailored approximations for the space–time integrals. The evaluation assesses goodness-of-fit using log-likelihood and AIC, and forecasting accuracy through MAE. Results are shown for both in–sample estimation of monthly counts and out–of–sample weekly predictions, as well as comparisons to time–series and stationary Hawkes baselines. The paper also inspects the learned spatial kernel and the fitted covariate coefficients to explain spatial heterogeneity in the intensity. Dong et al. (2024) analyze theft and robbery events in Valencia, focusing on incidents that occur along the city's street network. The model defines a self–exciting point process on the network, measuring spatial interaction by street–network distance rather than Euclidean distance, and using an exponential temporal kernel. To encode urban context, each event carries a mark that pairs the crime category with the functional zone inferred from nearby landmarks. The interactions among these marks are learned and regularized through a graph–attention architecture. Estimation proceeds by MLE using stochastic gradient descent over event subsequences. The study reports training and testing log–likelihood, AIC, and MAE for weekly event–count prediction, alongside qualitative maps of predicted network intensity. The learned mark–interaction structure is used to interpret how the type of crime and the surrounding landmark context relate to triggering patterns.

Both studies use self–exciting STPPs with MLE and gradient–based training, and both assess models using log–likelihood and AIC for fit and MAE for count prediction. The Atlanta study incorporates non-stationary spatial kernels with continuous-space covariates, whereas the Valencia analysis constrains interactions to street network topology and models crime-landmark relationships through learned mark interactions. Together, they show how modeling choices about space (continuous vs. network) and context (covariates vs. landmark–derived marks) shape evaluation under common metrics and provide interpretable summaries alongside predictive performance.

## 6.3 Traffic

Zhu et al. (2021a) model congestion events in Atlanta using an attention-based STPP that combines three components in the conditional intensity: a background term, an exogenous term linked to police 911 traffic incidents, and an endogenous self-excitation term learned by an attention mechanism. Directional spatial dependence is handled with a 'tail-up' formulation that uses stream distance on the highway network. Parameters are estimated by MLE with gradient-based optimization. The study reports goodness-of-fit with log-likelihood and assesses next-event prediction with location accuracy and MAE for event time, comparing to point-process and neural baselines on the same data.

Jin et al. (2023) represent the road system as a graph and integrates traffic states with congestion events in a spatiotemporal graph neural point process. Temporal context is encoded with a link-wise Transformer, spatial dependence with graph convolutions, and event history with a continuous GRU layer. Periodic effects, such as peak hours, are included through a gated term in the intensity function. The model jointly predicts the timing of the next congestion event and its duration. Training optimizes the NLL for inter-event times together with an absolute-error loss for duration. Evaluation on two city datasets uses NLL and MAE for occurrence time, and MAE for duration, alongside comparisons to graph-based and neural point-process baselines.

### 6.4 Epidemiology

Li et al. (2021b) treat confirmed COVID–19 cases as events and introduces an intensity–free generative STPP learned via imitation learning. The policy–based generator uses a recurrent history encoder and nonlinear transformations to sample the next event, bypassing direct likelihood evaluation and avoiding a fixed parametric intensity. Population, lockdown timing, and other covariates are incorporated as features to study how interventions may alter event dynamics. The evaluation focuses on recovering spatiotemporal patterns and forecasting next events, and includes uncertainty quantification for predicted case counts via Poisson confidence intervals. The study also contrasts the imitation–learning generator with an MLE–trained baseline and uses the learned model to explore counterfactual lockdown scenarios.

Dong et al. (2023c) analyze a COVID–19 dataset from Cali, Colombia that records the exact time and location of individual cases. They propose a non-stationary STPP in which the spatial triggering kernel varies by location and is parameterized through a simple neural mapping, allowing spatial anisotropy and heterogeneity while keeping the model interpretable. Exogenous effects from city landmarks (e.g., churches, schools, town halls) are added to the background intensity to capture setting–specific promotion. Parameters are learned by MLE with an efficient strategy for the likelihood integral. Model fit and forecasting are summarized with log–likelihood and MAE for one–week–ahead case counts, along with maps illustrating the predicted conditional intensity.

## 7 Open Challenges

### 7.1 Reproducibility

A significant barrier to advancing neural STPP research is the lack of standardized experimental setups and consistent baseline comparisons. In contrast to temporal point processes (TPPs), which benefit from unified libraries such as Xue et al. (2024), there is currently no widely adopted framework that jointly supports spatial and temporal modeling for STPPs. This limits the ability to incorporate and compare diverse neural architectures, evaluation metrics, and training strategies within a controlled environment, making rigorous ablation studies and reliable benchmarking more difficult.

Reproducibility (Bouthillier et al., 2019; Pham et al., 2020) in neural STPPs is particularly sensitive to issues arising from random sampling and stochastic training procedures. Variability induced by sampling in event sequence generation, Monte Carlo integration for likelihood estimation, and non-deterministic GPU operations (Wang et al., 2018) can introduce subtle discrepancies between runs. These effects may be amplified by the long-range temporal dependencies and coupled spatial-temporal dynamics of STPPs, where small variations in early predictions can propagate through subsequent events. Inconsistencies in data preprocessing, such as spatial coordinate normalization, event time quantization, or sub-sequence extraction strategies, can further complicate comparisons between studies.

Furthermore, most methods comprise multiple interchangeable components, such as different temporal encoders, spatial encoders, and interaction modeling mechanisms. Since new approaches often modify various components, it becomes difficult to attribute observed performance gains to a specific design choice. The interplay between modeling space, modeling time, and capturing spatiotemporal interaction effects introduces additional confounding factors, making it challenging to isolate the source of improvements.

A unified, open-source library with standardized data handling, reproducibility controls, modular architectural components, and configurable experimental protocols could facilitate more transparent, rigorous, and scalable research in neural STPPs.

## 7.2 Benchmarking

Understanding and evaluating new neural STPP methods requires standardized benchmarking datasets. Existing datasets often suffer from selection bias, missing data, and varying granularity, making integration and cross-study comparisons difficult. They also capture diverse spatiotemporal phenomena such as spatial heterogeneity, long-range dependencies, and entangled dynamics, with some showing self-exciting behavior and others self-correcting patterns. The lack of a unified, consistently tested dataset collection limits benchmarking to specific attributes or domains, while the absence of comprehensive contextual datasets hinders foundational model development and the study of event dynamics in data-scarce settings.

Key guidelines for dataset curation and splitting can facilitate fair and reproducible evaluation. Standardized schemas improve compatibility, with event-level data including timestamps, spatial coordinates, and optional categorical marks, alongside metadata specifying regions, time zones, and units. Covariates such as weather or temporal indicators can improve modeling, and having standard spatial and temporal resolutions enables meaningful comparison.

The literature points to several effective data-splitting strategies for testing generalization: temporal splits (e.g., roll-forward training on past events, testing on future periods), spatial splits (holding out entire regions), combined spatiotemporal splits (unseen regions and future periods), and cross-region splits (evaluating transferability across distinct geographic contexts). These approaches collectively offer a robust framework for assessing model performance across scenarios.

Synthetic datasets play an important role in neural STPP research by providing controlled environments for understanding model behavior under known conditions. Common examples include Pinwheel (Chen et al., 2021) for spatial heterogeneity and multimodal patterns, spatiotemporal Hawkes and self-correcting processes for self-exciting and self-correcting behaviors (Zhou et al., 2022), and HawkesGMM (Yuan et al., 2023) for cluster-based triggering with long-range dependencies. These benchmarks capture real-world phenomena such as entangled spatiotemporal dynamics, seasonal patterns, and varying background rates. Such controlled datasets allow for isolating specific model behaviors before applying methods to real-world data with unknown ground truth. We see a need for the creation of a simulation testbed for spatiotemporal point processes that captures key challenges in STPP modeling. The testbed should incorporate spatiotemporal entanglement, heterogeneous spatial and temporal patterns, multiple event types with interdependencies, and contextual effects that include both spurious and genuine causal predictors. Tunable parameters would allow the complexity of these factors to be varied, enabling controlled experiments across a range of modeling scenarios.

Real-world datasets in neural STPP research capture diverse spatiotemporal phenomena that challenge current methods. Earthquake catalogs from Japan (U.S. Geological Survey, 2020) and California (NCEDC, 2014; Ross et al., 2019) exhibit long-range dependencies with continuous processes and varying background rates. Epidemic datasets using COVID-19 data from New Jersey (Chen et al., 2021), ACLED conflict records, and EMPRES-i disease (Okawa et al., 2022) outbreak reports reveal self-exciting behavior with seasonal effects and entangled spatiotemporal dynamics. Urban mobility datasets, including Citibike NYC (Chen et al., 2021) and taxi records, capture spatial heterogeneity through high-volume flows with clustering patterns. Public safety datasets involving Atlanta 911 calls (Zhu et al., 2022), Chicago crime records (Okawa et al., 2019), Vancouver crime data, and NYC collision reports (Zhang et al., 2023) showcase both self-exciting and self-correcting patterns with heterogeneous event types across different urban environments.

Contextual datasets are becoming increasingly important as they provide environmental, demographic, and infrastructure information. For example, National Geophysical Data Center nightlights capture patterns of human activity, while Natural Earth land cover and GeoNetwork weather data explain spatial and temporal event variation. Additionally, population maps and gROADS contribute demographic and infrastructure context (Okawa et al., 2022). Many of these datasets now include multimodal information, such as text, categorical attributes, and numerical features, all aligned in both space and time. A unified repository that

links each event dataset to its associated contextual sources, annotated with geotags and collection periods, would be a valuable resource for advancing the field.

Building on this, the neural STPP literature points toward the broader need for standardized dataset repositories that combine synthetic benchmarks, real-world datasets, and contextual information under unified schemas. Such resources would feature consistent preprocessing pipelines, standardized data formats, and comprehensive documentation with versioned releases and fixed splits that support reproducible research.

### 7.3 Architectures

Generative strategies and alternative training objectives represent promising yet underexplored avenues for STPP research. While GAN-based architectures and Wasserstein objectives have shown promise in temporal point processes (Xiao et al., 2017), their adaptation to spatiotemporal applications remains limited. Similarly, methods moving beyond MLE, particularly those leveraging Transformer-based models instead of RNNs, could enhance the flexibility and robustness of event modeling. Continuous-time neural TPPs, such as those proposed by Biloš et al. (2021) and Chen et al. (2024), have demonstrated potential for temporal modeling but have yet to be sufficiently applied to spatial event modeling. Many existing neural STPPs treat space and time independently, missing opportunities to improve temporal predictions by capturing spatial relationships. Integrating GNNs to enhance spatial modeling remains an open challenge.

### 7.4 Interpretability

While interpretability in temporal point processes has received considerable attention, with advances in event attribution (Zhang et al., 2020b; Cüppers et al., 2024), counterfactual reasoning (Noorbakhsh and Rodriguez, 2022; Zhang et al., 2022; Großmann et al., 2024), and rule-based interpretability techniques (Li et al., 2021a; Kuang et al., 2024; Yang et al., 2024), research in the spatiotemporal domain remains relatively nascent. Extending interpretability to STPPs introduces unique challenges, such as disentangling spatial and temporal influences, capturing heterogeneous covariates, and accounting for interaction effects across modalities. Kernel-based methods provide a promising starting point, as they offer explicit representations of event-event relationships and triggering patterns. However, they often struggle to model background rates unrelated to observed events, suggesting opportunities for hybrid formulations. Recent progress in deep kernel (Dong et al., 2023b; Zhu et al., 2022) approaches and kernel-dependent designs indicates that it is possible to retain interpretability while improving flexibility.

Recent works (Okawa et al., 2022; 2019; Dong et al., 2024) have explored the use of covariates, such as satellite imagery and local landmarks, to inform event rates and propagation dynamics. These approaches leverage contextual data to retain interpretability through kernel-based formulations, while offering greater flexibility by moving beyond standard parametric kernel assumptions. This research direction remains promising and warrants further investigation, particularly by extending beyond parametric forms. Potential avenues include continuous-time formulations combined with covariates to reveal how specific factors shape event rates and propagation over space and time without discretization artifacts, as well as fully neural or diffusion-based methods to more deeply investigate the effects of contextual variables.

Semi-parametric models (Zhou et al., 2022; Zhou and Yu, 2023) continue to offer valuable interpretability by decomposing the conditional intensity into components such as baseline rates, self and cross excitation, covariate effects, and spatial terms. Developing STPP models with semi-parametric components is a promising direction of research as it retains flexibility while also improving usability through interpretable components.

Reinforcement learning-based approaches have been studied in the context of temporal point processes (Li et al., 2018; Upadhyay et al., 2018; Qu et al., 2023); however, only a few works have extended them to the spatiotemporal setting (Zhu et al., 2021b; Li et al., 2021b), which remains relatively understudied. This is a promising avenue of research, as RL formulations learn policies that capture event dynamics, while counterfactual rollouts offer a way to examine the effects of different interventions and their potential consequences.

For fully neural STPPs, a major opportunity lies in developing techniques that expose the decision process of the model without significantly sacrificing performance. Possible directions include explicit intensity decom-

positions (Panos, 2024) within neural architectures, attention mechanisms capturing covariate influence and event-event relationships (Meng et al., 2024a; Shou et al., 2023), interpretable latent representations (Meng et al., 2024b), and counterfactual generators that systematically probe the impact of spatial or temporal perturbations. Addressing these challenges will not only improve transparency but also enable the use of neural STPPs in sensitive domains where explainability is essential for trust and adoption.

### 7.5 Applicability

Despite improvements in predictive performance, neural STPPs are rarely utilized to inform policy decisions or design interventions. Their lack of interpretability limits their applicability in real-world scenarios, where generalizable, interpretable models with uncertainty quantification are essential. Policy-makers often seek to understand the impact of interventions, necessitating models that incorporate contextual factors and causal effects in event propagation. This underscores the need for research in interpretable neural networks, neuro-symbolic methods, and counterfactual analysis for retrospective policy evaluation and deeper insights into event modeling.

### 7.6 Causality and Uncertainty Quantification

Policy-focused applications of neural STPPs require deeper insights into event propagation and the causal factors influencing event rates. However, causal inference and uncertainty quantification remain underexplored in this field. There has been notable progress in causal inference for neural spatiotemporal methods (Oprescu et al., 2025; Wang et al., 2024b; Deng et al., 2023) and for spatiotemporal point processes (Papadogeorgou et al., 2022; Christiansen et al., 2022), but the extension of these ideas to *neural* STPPs has been limited. Most existing neural STPPs optimize for predictive likelihood or simulation fidelity, capturing statistical dependencies in space and time but without explicitly distinguishing correlation from causation. In many real-world domains, such as modeling disease spread, human mobility, or crime policy decisions rely on determining whether an observed influence reflects a genuine causal relationship or shared exogenous drivers. For example, a model may learn that events in one region often precede events in a neighboring area, but without causal analysis, it is unclear whether this is due to true propagation, common underlying conditions, or artifacts of data collection. Causal representation learning for STPPs must address spatio-temporal confounding (Wang et al., 2024a), and existing advances in causal inference for temporal point processes (Gong et al., 2024a; Gao et al., 2021; Zhang et al., 2022; Noorbakhsh and Rodriguez, 2022) could be extended to the spatiotemporal domain, potentially leveraging structured intensity decompositions, counterfactual reasoning, and invariant representation learning (Arjovsky et al., 2019).

Uncertainty quantification is equally important for neural STPPs, particularly in policy-facing settings where overconfident predictions can have harmful consequences. Unlike traditional statistical STPPs, where uncertainty is often explicit in the parametric form, many neural models produce point estimates of intensity functions or event parameters without calibrated confidence measures. Bayesian methods, which could integrate expert knowledge through priors and yield posterior uncertainty estimates, have not been widely adopted in neural spatiotemporal event modeling (Dubey et al., 2023a). Other promising directions include variational inference for intensity distributions (Li et al., 2024) and deep ensembles (Dubey et al., 2023b; Rahaman et al., 2021) to capture epistemic uncertainty. In spatiotemporal contexts, structured uncertainty modeling could further account for heteroskedasticity arising from uneven spatial coverage, time-varying data quality, or rare-event regimes.

Addressing causality and uncertainty jointly is particularly important. Causal effect estimates without uncertainty quantification risk overconfident and misleading recommendations, while uncertainty estimates without causal grounding cannot separate genuine effect variability from confounding. Developing neural STPP frameworks that integrate causal structure learning with calibrated uncertainty, such as Bayesian causal neural architectures or counterfactual generative models, could lead to robust, interpretable, and trustworthy models for complex spatio-temporal phenomena, enabling better-informed decision-making and intervention design.

## Broader Impact

Point processes, including their spatiotemporal extensions, are increasingly applied in ethically sensitive domains such as police patrol allocation (Mohler et al., 2015), epidemiological surveillance (Alikhademi et al., 2022), disaster response, and urban resource planning. In such settings, the stakes of predictive modeling are high, and errors or biases in predictions may cause disparate harms across demographic, socioeconomic, or geographic groups. For example, crime prediction models have been shown to risk reinforcing existing structural biases when trained on historically biased datasets, leading to potential over-policing of certain communities (Lum and Isaac, 2016; Ensign et al., 2018; Brantingham et al., 2018). Similarly, in public health contexts, biases in surveillance data may exacerbate disparities in detection and intervention across populations.

The use of neural STPPs for spatial interventions introduces specific ethical challenges. First, data-related issues (Westreich, 2012; Price and Ball, 2015) such as selection bias, missing data, under-reporting, or measurement error can lead to distorted inferences. While such issues affect both traditional statistical STPPs and their neural counterparts, the latter may be more difficult to audit due to their complexity and lack of interpretability. As a result, it is important to critically assess whether these models can be relied upon to inform policy decisions (Dong and Xie, 2024), particularly in high-impact settings where predictive outputs might be misinterpreted as causal.

Second, contextual covariates used in modeling, such as satellite imagery, survey-derived socioeconomic indicators, or night-time light intensity, may act as proxies for sensitive attributes like race, wealth, or ethnicity (Weidmann and Theunissen, 2021; Mellander et al., 2015; Johnson et al., 2022). When combined with event data, such proxies can inadvertently enable models to leverage sensitive attributes, leading to predictions that embed or amplify societal biases. The indirect nature of proxy variables makes these risks harder to detect, necessitating careful auditing of feature relevance and its implications for fairness.

Third, there are important privacy considerations (Cao et al., 2019). Spatiotemporal event data, especially when linked to individuals or small communities, can be highly sensitive. Inference on fine-grained temporal or spatial patterns may inadvertently reveal private information, even when data is aggregated or anonymized. This raises the need for privacy-preserving modeling approaches, such as differential privacy (Dwork et al., 2014), secure multi-party computation, or federated learning (Gong et al., 2024b; Wang et al., 2024c), when deploying neural STPPs in real-world scenarios.

Finally, governance and oversight frameworks (European Union, 2024; Dafoe, 2018) are often lacking in the deployment of predictive spatiotemporal models. Without clear accountability structures, it can be difficult to ensure that model outputs are used responsibly and that affected communities have a voice in the design, validation, and deployment of these systems.

To mitigate these risks, we advocate the following strategies: (i) fairness-aware evaluation metrics (Shang et al., 2020; Mitchell et al., 2021) to measure and correct disparate impacts across subgroups; (ii) bias detection and de-biasing techniques (Mohler et al., 2018; 2019; Li et al., 2023) in training data and model outputs; (iii) interpretability tools (subsection 7.4) for neural STPPs, such as intensity decomposition and event attribution, to allow auditing of model reasoning; (iv) transparency in reporting data provenance, preprocessing choices, and model limitations; and (v) engagement with domain experts, policymakers, and stakeholders to align modeling objectives with ethical and societal goals.

Ultimately, the potential benefits of neural STPPs, such as improved predictive performance and richer modeling of complex spatiotemporal dynamics, must be weighed carefully against the ethical risks. Their deployment should be accompanied by rigorous bias analysis, privacy safeguards, and governance mechanisms to ensure that these methods contribute to equitable and responsible decision-making.

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
