# OpenReview forum: "Neural Spatiotemporal Point Processes: Trends and Challenges"
_TMLR — Accepted by TMLR_

### Review · Reviewer_dApF · 2025-07-14

**Summary Of Contributions:**

The submission provides a comprehensive review of parameter estimation and inference methods for spatiotemporal point processes (STPPs), focusing on neural network-based approaches. Maximum likelihood, score matching, and reinforcement learning techniques for parameter estimation. Task-specific evaluation metrics for assessing STPP performance, including distributional fit, prediction accuracy, and uncertainty quantification. Real-world applications in natural disasters, crime, traffic, and epidemiology. Open challenges including reproducibility, benchmarking datasets, computational costs, and interpretability.

Strengths:
- Comprehensive coverage of both classical and modern inference techniques, including cutting-edge neural and non-parametric approaches.
- Balanced discussion on evaluation metrics, emphasizing practical trade-offs between likelihood-based and distributional measures.
- Real-world relevance is emphasized via application sections grounded in diverse domains.
- Identifies reproducibility and benchmarking as key gaps, showing awareness of broader research needs.

Limitations:
- While it mentions challenges like interpretability, the review does not deeply engage with specific methods to address these issues.
- Heavy reliance on recent preprints and workshop papers (e.g., Lüdke et al., 2025) suggests some cited methods may not yet be widely validated.
- The text is dense and assumes the reader already has foundational knowledge of STPPs and neural methods, which may limit accessibility.
- There is less focus on causal inference or fairness aspects, both of which are increasingly relevant in real-world STPP applications.

**Additional Comments:**

boderline reject

**Audience:**

Yes

**Audience Explanation:**

TMLR readers interested in time series, structured prediction, or generative models would find it relevant. The paper discusses neural point processes, estimation methods, evaluation challenges, and reproducibility, all key concerns in machine learning research and practice.

**Broader Impact Concerns:**

- Crime prediction models and epidemiological surveillance carry clear ethical risks: privacy violations, reinforcing social biases, or misinterpretation of predictive outputs as causal.
- The review does not explicitly address data privacy, algorithmic bias, or governance frameworks, which is increasingly expected for works in this area.
- Suggested broader impact section additions: discuss risk mitigation strategies, such as privacy-preserving learning or fairness-aware evaluation metrics.

**Claims And Evidence:**

Yes

**Claims Explanation:**

- The submission systematically references recent literature and explicitly notes trade-offs between methods (e.g., computational complexity vs. flexibility).
- Metrics and applications are tied back to specific studies, providing concrete examples.
- Some claims would benefit from clearer quantitative comparisons or reproducible benchmarks, which the authors themselves acknowledge as a limitation.

**Requested Changes:**

- Clarify and expand on interpretability strategies for neural STPPs beyond just identifying it as a challenge (e.g., explainability techniques, causal discovery).
- Provide a table or structured list summarizing the strengths and weaknesses of each inference method discussed (MLE, score matching, RL).
- Include a more concrete proposal or roadmap for unified benchmarking datasets, what domains, what characteristics, potential public data sources?
- Recheck consistency in citations, ensure all mentioned works are properly referenced, especially newer or unpublished studies like Lüdke et al., 2025.
- Lightly simplify technical explanations where possible to make the review accessible to a broader audience.

---

> ### Author Response · Authors · 2025-08-18
> **Response to Reviewer dApF**
>
> The authors thank Reviewer dApF for the constructive and detailed feedback, which has been very helpful in strengthening the paper. The concerns raised have been carefully addressed in the revised manuscript.
>
>
> **Interpretability (Section 7.4).** A dedicated subsection on interpretability has been added. This section now moves beyond simply identifying interpretability as a challenge and engages with specific methods that could be applied in neural STPPs. These include attention-based analysis, counterfactual reasoning, and neurosymbolic approaches, as well as recent work on interpretability in neural temporal point processes and preliminary contributions in neural STPPs. Potential future directions are outlined, with a strong grounding in the unique properties of spatiotemporal point processes.
>
>
> **Causality and Uncertainty (Section 7.6).** The discussion of causality and uncertainty has been expanded to connect these directions more directly to the characteristics of spatiotemporal data. Recent advances in causal inference for neural spatiotemporal systems are reviewed alongside standard causal approaches from the STPP literature. In addition, methods developed for temporal point processes are considered in terms of their extension to neural STPPs. The section also highlights recent efforts on uncertainty quantification, outlining both progress and open challenges. The revised discussion emphasises that causality and uncertainty are coupled research directions that must be addressed together for practical applications of neural STPPs.
>
>
> **Inference Methods (Section 4).** To clarify the trade-offs between inference methods, a comparative table has been introduced. The table contrasts Maximum Likelihood, Amortised Variational Inference, Score Matching, and Reinforcement/Imitation Learning, summarising the key ideas behind each approach and outlining their respective strengths and limitations. This addition improves accessibility for readers who may be less familiar with the inference literature.
>
> **Benchmarking Roadmap (Section 7.1).** In response to the reviewer’s request for a more concrete proposal, a structured benchmarking framework has been added. The framework specifies a minimal event schema (time, location, event marks, and metadata such as region, timezone, and units) and proposes standardised evaluation protocols such as temporal roll-forward validation, spatial holdouts, spatiotemporal splits, and cross-region transfer testing. The roadmap integrates both synthetic datasets, for controlled testing of specific mechanisms, and real-world datasets from domains such as seismology, epidemiology, human mobility, and crime. These datasets, drawn from public repositories, were selected for their diversity and enriched with contextual information.
>
>
> **Citations and Accessibility.** The reviewer noted the reliance on recent preprints and workshop papers. In the revision, such works are retained only when they represent important emerging contributions and are clearly marked as preliminary. At the same time, the literature review has been broadened to include more established references for balance. To further improve accessibility, the text has been revised with brief summaries at the start of technical sections, clarified notation, and new comparative tables to make the survey easier to follow for readers without a deep prior background in neural STPPs.
>
>
> **Broader Impact.** The previous ethical statement has been replaced with a substantially expanded broader impact section. Beyond issues such as bias and missing data, the revised discussion now addresses fairness, privacy, interpretability, and governance frameworks. Specific risks in sensitive domains such as crime prediction and epidemiological surveillance are considered, including privacy violations, algorithmic feedback loops that may reinforce social biases, and the danger of misinterpreting predictive outputs as causal. Relevant work on privacy-preserving learning and fairness-aware evaluation is cited, and future research directions are outlined.
>
>
> Overall, the revisions respond directly to the reviewer’s feedback. By deepening the engagement with interpretability, causality, fairness, and accessibility, while also clarifying trade-offs and adding a concrete benchmarking roadmap, the paper now offers a more balanced, rigorous, and usable survey of neural STPPs.

---

### Review · Reviewer_M8P1 · 2025-07-23

**Summary Of Contributions:**

### Summary.
This work reviews modern spatio-temporal point processes (STPPs) that leverage deep neural networks to model the intensity function and discusses the open challenges that remain for advancing the development of STPPs. Specifically, Section 3 introduces methods for modeling the intensity function, including history event-based encoders, kernel functions, and continuous-time normalizing flows for single-event prediction, as well as transformer-based normalizing flows or diffusion models for multi-event prediction. Section 4 then describes parameter estimation methods, including maximum likelihood estimation (MLE), amortized inference, score matching, and reinforcement learning. Section 5 presents the evaluation metrics for the trained models, followed by applications of STPPs in Section 6. Finally, Section 7 discusses open challenges in this area.

### Contribution.

The main contribution of this work is (1) to categorize deep neural network-based spatio-temporal point processes (STPPs) from a modeling perspective, and (2) to present future research directions necessary for advancing the development of STPPs.

**Additional Comments:**

N/A

**Audience:**

Yes

**Audience Explanation:**

To the best of my knowledge, review papers on deep neural network-based spatio-temporal point processes (STPPs) are still limited, with the exception of [1]. While this prior work overlaps in its discussion of kernel methods, the present paper offers a broader perspective. In particular, if this work were to more explicitly introduce continuous time-based methods, such as continuous normalizing flows or diffusion models for STPPs, it could be of significant interest to readers seeking to understand recent trends in generative approaches to STPP modeling.

[1] Deep Spatio-temporal Point Processes: Advances and New Directions – arXiv

**Claims And Evidence:**

No

**Claims Explanation:**

This work is written as a review of recent trends in spatio-temporal point processes (STPPs), but it lacks detailed descriptions of methods and empirical results to support the authors' claims. As a result, my comments are based solely on the authors’ categorization and high-level assertions.

> Regarding the categorization presented in Section 3, the current description primarily enumerates existing methods based on modeling techniques. However, it is unclear why certain transitions from older to newer approaches are considered. For instance, it is not evident why the kernel method is introduced instead of using a history-based encoder, or what specific advantages the kernel method offers. Clarifying the motivations and benefits behind each modeling choice would help readers better understand not only the modeling techniques themselves but also the goals and advantages that guide their development.

> In addition, the criteria for categorization seem unclear. Section 3.1 appears to be organized by architectural choices, whereas Sections 3.2 and 3.3 are categorized by task type. This inconsistency may confuse readers about the overall structure and intention behind the categorization.

> As for the open challenges discussed in Section 7, the rationale for each point tends to be presented from a general machine learning perspective rather than being grounded in the specific context of STPPs. As a result, the suggested future research directions come across as overly broad.

> In particular, Section 7.4 on Applicability states that neural STPPs are rarely used to inform policy decisions or design interventions. The authors attribute this mainly to the lack of interpretability and therefore suggest studying "interpretable neural networks, neurosymbolic methods, and counterfactual analysis for retrospective policy evaluation." However, this direction seems misaligned. To me, it would be more appropriate to emphasize the need for improving the interpretability of STPPs themselves, rather than broadly targeting applicability as the main future direction.

> Similarly, in Section 7.5, the authors argue that causality and uncertainty in STPPs have been underexplored and should therefore be studied. While this is a reasonable claim, the justification feels too generic and insufficiently tied to the unique characteristics of STPPs. Strengthening this section by explicitly connecting these research directions to the structural and functional properties of STPPs would make the argument more compelling.

**Requested Changes:**

* Regarding Section 3, could the authors clarify the reasoning behind the proposed categorization? Additionally, would it be possible to include an “umbrella-style” illustration that visually summarizes the categorization of STPPs? Such a figure could help readers better understand the overall structure and relationships between the different modeling approaches.

* In Section 3, could the authors elaborate on the advantages and objectives of each modeling method? It would also be helpful to explain the motivations for the transitions between modeling paradigms—e.g., why certain methods (e.g., kernel-based approaches) are introduced as alternatives to previous techniques (e.g., history-based encoders), and what specific benefits they offer.

*  For Section 7, could the authors provide more concrete justifications for the importance of addressing each open challenge? Connecting these challenges more explicitly to the unique characteristics and limitations of STPPs would help readers appreciate why they are pressing research directions, rather than general challenges in machine learning.

---

> ### Author Response · Authors · 2025-08-18
> **Response to Reviewer M8P1**
>
> The authors thank Reviewer M8P1 for the insightful comments and constructive feedback, which have helped refine both the structure and depth of the survey. The key revisions are as follows:
>
>
> **Section 3 (Categorisation and Diagram).** The rationale behind the categorisation of modelling approaches is now stated explicitly in the header paragraph. To further aid readers, a taxonomy diagram has been added to visually summarise the categories and illustrate their relationships. The section has also been restructured for improved flow: each subsection begins with an explanation of how the approach differs from those presented earlier, highlighting the specific benefits and trade-offs. The discussion of continuous-time methods and diffusion-based approaches has been expanded to show how events are modelled in these settings, and the coverage of multi-event prediction methods has been extended to clarify their differences from single-event approaches. Together, these revisions ensure that the categorisation is both coherent and well-motivated, addressing the reviewer’s concern about consistency and clarity.
>
>
> **Section 7 (Open Challenges).** All subsections in Section 7 have been expanded to tie the challenges more directly to issues specific to neural STPPs. For example, in Section 7.1, concrete issues regarding reproducibility are described, and a roadmap for a benchmarking suite is introduced. This proposal includes synthetic environments to test core mechanisms under controlled conditions, as well as protocols for spatiotemporal evaluation that address the particular difficulties of STPP research. These additions respond to the reviewer’s request for more grounded and concrete discussion of open challenges in neural STPPs, rather than general issues in machine learning research.
>
>
> **Interpretability (new subsection).** A dedicated subsection on interpretability has been added. This section discusses recent progress in neural temporal point processes, highlights preliminary work on neural STPPs, and outlines potential future directions. In line with the reviewer’s suggestion, it also proposes ways to adapt interpretability techniques from other areas of machine learning, such as attention analysis and counterfactual reasoning, while keeping the discussion grounded in the spatiotemporal point process setting.
>
>
> **Section 7.6 (Causality and Uncertainty, formerly Section 7.5).** This section has been substantially expanded to ground the discussion in the unique characteristics of spatiotemporal data and neural STPPs. The revision cites recent work on causal inference in neural spatiotemporal systems, contributions from the classical STPP literature, and advances in neural temporal point processes. The section also expands on uncertainty quantification, citing recent efforts and outlining directions for future work. The argument has been reframed to emphasise that causality and uncertainty are coupled research directions, and that progress on both is essential for the practical deployment of neural STPPs.
>
>
> Overall, these revisions improve the clarity, motivation, and coherence of the survey. By strengthening the categorisation of methods, providing a visual taxonomy, introducing a focused discussion on interpretability, and grounding open challenges more concretely in the specific problems of neural STPPs, the revised manuscript directly addresses the reviewer’s concerns.

---

### Review · Reviewer_PUWP · 2025-08-11

**Summary Of Contributions:**

The authors provide, to the best of my knowledge, the first survey on neural spatiotemporal point processes (STPPs). Neural STPPs have been a research area of great interest in the ML community in recent years, and having a comprehensive survey of recent developments would be a benefit to the community.

## Strengths
- The survey is quite broad, covering most of the key literature in the area that I am aware of.
- It is well organized, divided into sections on modeling, parameter estimation, evaluation, and applications.

## Weaknesses
Several of these sections are lacking in depth, with the discussion looking more like what is found in the related work section for a methods paper, rather than a survey. See my requested changes for more details.

**Additional Comments:**

- In Section 2.1, the last sentence refers the reader to Figure 1, which illustrates the $t_i$'s and $\\mathbf{s}_i$'s, but not the history $\\mathcal{H}_t$, which should be added.
- Needs a thorough proofread. Lots of minor typos throughout, including the following in just the first few pages:
  - Section 2.3, second sentence: add space between "the" and $\\mathcal{H}_t$.
  - Section 2.3, last sentence is missing a period.
  - Section 3, second paragraph: Capitalize first word in first sentence.
  - Vaswani et al. (2017) reference for "Attention is all you need" is missing the other authors.
  - All equations have the word "equation" duplicated.
- Table 1 should probably be moved to a later page, closer to Section 5.

**Audience:**

Yes

**Audience Explanation:**

Yes, as there is a large body of ML research on neural STPPs.

**Broader Impact Concerns:**

There is a brief ethical statement that should probably be expanded. The authors discuss data-related issues such as selection bias, missing data, etc. in Section 7.2. There is no discussion on how neural STPPs compare to traditional statistical STPPs in the presence of such issues, and whether they can be trusted for drawing conclusions, e.g., to inform policy decisions.

**Claims And Evidence:**

No

**Claims Explanation:**

The claims are partially supported--the authors claim to provide comprehensive coverage on neural STPPs, and they succeed in terms of breadth, but several sections are lacking depth.

**Requested Changes:**

- Section 3.2.2 on continuous-time normalizing flow (CNF)-based methods is extremely short and lacking in detail. Given the long and detailed description on kernel-based methods in Section 3.2.1, I would suggest adding a few paragraphs and perhaps a figure or equations illustrating the difference between these CNF and kernel-based methods.
- Sections 3.3.1 and 3.3.2 are also lacking in depth. The advantages of multi-event prediction models are described. What are the disadvantages?
- Last sentence of Section 5: provide some references or justification as to why recent work favors WD (What is WD? Wasserstein distance?) and MMD for assessing generative quality.
- Section 6 on applications is also lacking in depth. Beyond just providing a citation to the application paper, it would be useful to provide some type of information (either in the body text or in the form of a table) on how the different studies relate in the methods they used--what model, what estimation approach, and what evaluation metric.

---

> ### Author Response · Authors · 2025-08-18
> **Response to Reviewer PUWP**
>
> The authors thank Reviewer PUWP for the detailed feedback and constructive suggestions, which have helped improve both the clarity and depth of the paper. The revisions made in response are summarised below.
>
> **Section 3.2.2 (CNFs and Neural ODEs).** This section has been substantially expanded. It now includes a detailed description of how events are modelled using Neural ODEs and Continuous Normalising Flows (CNFs). Equations have been added to illustrate how the latent continuous hidden state is updated and how CNFs model the spatial distribution of events in a non-parametric manner, which is fundamentally different from kernel-based approaches. A new introductory paragraph clarifies the motivation for adopting CNFs over kernels, and the discussion has been expanded to include history conditioning and time-varying CNFs, thereby addressing the reviewer’s request for greater depth and contrast with kernel-based methods.
>
>
> **Sections 3.3.1 and 3.3.2 (Multi-event prediction).** These sections have been expanded to provide a clearer comparison with single-event prediction approaches. In addition to discussing the advantages of multi-event parameterisations, the revision now explicitly lists their disadvantages, giving a more balanced treatment of the trade-offs involved.
>
>
> **Section 5 (Evaluation metrics).** The final part of this section has been elaborated to provide detailed justification and references for the use of Wasserstein distance as a preferred evaluation metric in certain contexts. A comparison with negative log-likelihood (NLL) is also included, highlighting situations where NLL may be more appropriate, thereby clarifying the complementary roles of these measures.
>
>
> **Section 6 (Applications).** This section has been expanded to go beyond citation and provide context about the cited studies. Specifically, details have been added on the models used, estimation approaches, and evaluation metrics. This comparative perspective helps readers better understand how the methods relate to one another across application domains.
>
>
> **Broader Impact (Section 7.2).** The previous brief ethical statement has been replaced with a substantially expanded, broader impact section. This now discusses not only data-related issues such as bias and missing data, but also broader concerns around fairness, privacy, interpretability, and the implications of applying neural STPPs in high-stakes contexts such as policy-making. Governance frameworks are also referenced to highlight responsible pathways for real-world deployment.
>
> **Figure 1.** The figure is intended to depict three events as representing the history, and for this reason, no additional history symbol was added.
>
> **Proofreading.** The manuscript has been carefully proofread, and the errors noted by the reviewer have been corrected.
>
> **Table 1. relocation** The former Table 1 (now Table 2) has been relocated closer to Section 5, as suggested, to improve the flow of the manuscript.
>
>
> Overall, the revisions directly address the reviewer’s comments. By expanding the discussion of CNFs and Neural ODEs, clarifying the trade-offs in multi-event prediction, strengthening the treatment of evaluation metrics and applications, and broadening the ethical discussion into a comprehensive broader impact section, the revised manuscript now provides a clearer, deeper, and more balanced survey of neural STPPs.

---

### Comment · Editors_In_Chief · 2025-12-15

On December 15, 2025, at the request of the authors, a new camera-ready version was uploaded by the Editors in Chief. The authors state that the only differences are in the citation format (i.e., correct usage of \citep and \citet) and some typos in the references.

---

### Decision · Action_Editor_sBUu · 2025-10-02

**Recommendation:** Accept with minor revision

**Additional Comments:**

The reviewers found that the paper has improved significantly since the first submitted version, but that there are still a few points that should be addressed in a minor revision (quotes from reviewers below):

- Table 1 is too wide and running into the margin. I think it would still look fine if narrowed to the text width.

- Section 7.2: "We propose the creation of a simulation testbed..." This reads as if you proposed the simulation testbed in your paper, which I do not see. Instead, I interpret this as a call to the community to create such a testbed. This could be rephrased to make it clear, e.g., "We see a need for creation of a simulation testbed..."

Overall, I feel that the contents of the manuscript have improved compared to the previous version. However, it still seems to lack clarity because the authors present too much information without sufficiently considering the reader’s understanding up to that paragraph. This writing makes it difficult for readers to maintain a clear sense of flow.

For example, after reading the introductory part of Section 3, I lost the main point because I expected a clear outline of the contents in Sections 3.1, 3.2, and 3.3. Instead, additional information, beyond what was anticipated, was presented in an overly detailed manner. The following is written at the beginning of Section 3, and the third paragraph below illustrates this issue:

> Our categorisation in this section reflects both architectural and task-specific considerations. Section 3.1 covers history event encoders, presented as architectural components that can be paired with a variety of decoding strategies. These encoders implemented via RNNs, Transformers, or other architectures serve as reusable mechanisms for representing past event sequences, and can be applied in both single-event and multi-event prediction settings

> In contrast, Sections 3.2 and 3.3 are organised by prediction task. For single-event prediction, we include only modelling paradigms that are specific to this task namely, kernel-based approaches, continuous-time neural ODE/CNF models and non-parametric generative models which require sequential likelihood evaluation for each predicted event. For multi-event prediction, we include only decoders that are explicitly designed to generate multiple events or predict entire event sequences; these may use any history encoding scheme or latent modelling approach, but their decoding objective is sequence-level rather than event-by-event

> We begin with spatial encoding strategies, as they are a core challenge in STPPs and strongly influence model performance, before moving to the task-specific modelling paradigms. Temporal encoding methods are not a primary focus here, as they are already well covered in prior surveys Shchur et al. (2021). Modelling choices are often driven by the application context. For example, crime forecasting may prioritise modelling the influence of specific landmarks, earthquake prediction may require fine-grained spatial accuracy, and epidemic modelling might emphasise regional policy effects, favouring coarser spatial resolution. Some applications impose parametric constraints to enable hypothesis testing, while safety-critical settings may require explicit uncertainty quantification. These considerations shape the choice of spatial encoding strategy.

We would like to invite the authors to submit a revised camera-ready version that addresses those comments.

**Audience:**

Yes

**Audience Explanation:**

The reviewers found that it is a robust, high-quality survey that would help readers learn about this research field.

**Claims And Evidence:**

Yes

**Claims Explanation:**

The paper is a survey paper that provides an overview of neural spatiotemporal point processes. The reviewers found that it covers the breadth of the field sufficiently.